# The Effects of Mechanical Load on Chondrogenic Responses of Bone Marrow Mesenchymal Stem Cells and Chondrocytes Encapsulated in Chondroitin Sulfate-Based Hydrogel

**DOI:** 10.3390/ijms24032915

**Published:** 2023-02-02

**Authors:** Ilona Uzieliene, Daiva Bironaite, Edvardas Bagdonas, Jolita Pachaleva, Arkadij Sobolev, Wei-Bor Tsai, Giedrius Kvederas, Eiva Bernotiene

**Affiliations:** 1Department of Regenerative Medicine, State Research Institute Centre for Innovative Medicine, LT-08406 Vilnius, Lithuania; 2Latvian Institute of Organic Synthesis, LV-1006 Riga, Latvia; 3Department of Chemical Engineering, National Taiwan University, Taipei 104, Taiwan; 4The Clinic of Rheumatology, Orthopaedics Traumatology and Reconstructive Surgery, Institute of Clinical Medicine, Faculty of Medicine, Vilnius University, LT-03101 Vilnius, Lithuania

**Keywords:** bone marrow mesenchymal stem cells, chondrocytes, chondroitin sulfate tyramine hydrogels, chondrogenic differentiation, mechanical compression/load, cartilage explants, cartilage regeneration

## Abstract

Articular cartilage is vulnerable to mechanical overload and has limited ability to restore lesions, which leads to the development of chronic diseases such as osteoarthritis (OA). In this study, the chondrogenic responses of human bone marrow mesenchymal stem cells (BMMSCs) and OA cartilage-derived chondrocytes in 3D chondroitin sulfate-tyramine/gelatin (CS-Tyr)/Gel) hydrogels with or without experimental mechanical load have been investigated. Chondrocytes were smaller in size, had slower proliferation rate and higher level of intracellular calcium (iCa^2+^) compared to BMMSCs. Under 3D chondrogenic conditions in CS-Tyr/Gel with or without TGF-β3, chondrocytes more intensively secreted cartilage oligomeric matrix protein (COMP) and expressed collagen type II (*COL2A1*) and aggrecan (*ACAN*) genes but were more susceptible to mechanical load compared to BMMSCs. ICa^2+^ was more stably controlled in CS-Tyr/Gel/BMMSCs than in CS-Tyr/Gel/chondrocytes ones, through the expression of L-type channel subunit CaV1.2 (*CACNA1C*) and Serca2 pump (*ATP2A2*) genes, and their balance was kept more stable. Due to the lower susceptibility to mechanical load, BMMSCs in CS-Tyr/Gel hydrogel may have an advantage over chondrocytes in application for cartilage regeneration purposes. The mechanical overload related cartilage damage in vivo and the vague regenerative processes of OA chondrocytes might be associated to the inefficient control of iCa^2+^ regulating channels.

## 1. Introduction

Human articular cartilage plays an essential role in cushioning joint pressure during motion and reducing bone friction. The avascular nature of cartilage shortens the nutrient supply to chondrocytes, making cartilage tissue more vulnerable and fragile [1]. The various types of strong physical forces, such as mechanical compression or overload, tension, shear stress or hydrostatic pressure applied to the cartilage might cause deformation of extracellular matrix (ECM) and impair chondrocyte functioning [2,3]. Therefore, the investigation of cartilage-tissue-damaging mechanisms under experimental mechanical load conditions in vitro will give important additional information about targeted cartilage regeneration in vivo.

Chondrocytes are not capable of fully restoring cartilage tissue, particularly after trauma or during degenerative diseases such as osteoarthritis (OA) [4]. To overcome these issues, tissue engineering is providing possibilities to investigate cartilage regeneration mechanisms, applying cells and hydrogels that could increase chondrogenic differentiation in vitro, furthering their therapeutic applications in vivo [5]. It was shown that adult tissue-derived stem cells have a highest tissue origin specificity and are more prone to differentiate into tissues from where they are derived [6]. Chondrocytes, which comprise the only cellular component of articular cartilage, arise from the mesenchymal progenitor cells during skeletal development, synthesize cartilage forming compounds and better stimulate cartilage regenerating properties compared to the other stem cells [7]. However, since the availability of healthy chondrocytes is limited, and OA cartilage-derived chondrocytes are characterized by an altered intracellular signaling leading to the limited regenerative potential [8], various types of adult tissue-derived mesenchymal stem cells (MSCs), including umbilical cord, adipose, synovium tissue-derived and others, are of interest as a promising cell source for cartilage tissue regeneration [9,10]. Human bone marrow mesenchymal stem cells (BMMSCs), due to their greater chondrogenic potential and easier availability compared to the other adult tissue-derived MSCs, have higher cartilage regenerating applicability both in 2D and 3D model systems [5,10,11]. Which type of cells, BMMSCs or cartilage-derived chondrocytes is more suitable for cartilage engineering purposes using injectable hydrogels under mechanical load is not clear.

Three-dimensional natural, chondrogenic origin-based materials are in the lead to mimic chondrogenic differentiation conditions in vivo [12,13]. Chondroitin sulfate (CS) is a glycosaminoglycan (GAG) most abundant in the cartilage and can be a perfect component for cartilage regeneration studies both in vitro and in vivo [14]. Besides the natural components mimicking 3D cell environment in cartilage, mechanical load is an integral part of model systems investigating cartilage regeneration in vitro [15]. However, mechanical load, depending on the intensity and duration, could activate different molecular mechanisms, positively or negatively affecting cell viability, growth and chondrogenic phenotype [16,17]. If mechanical load leads to downregulation of cartilage specific ECM components, it is too strong and should be considered as a mechanical overload, while mechanical load stimulating cartilage-specific proteins and genes can be named as a mild load. The 3D dynamic culturing of canine chondrocytes on scaffolds of various topology were shown to better support their viability than static conditions [18] and the mild mechanical stimulation of various scaffold-cell composites or bioreactors stimulated chondrogenesis by evaluating the synthesis of transcription factors, collagen II and ECM components [19,20]. However, how the strong experimental mechanical load affects chondrogenic differentiation of BMMSCs and OA-derived chondrocytes in hydrogels and/or cartilage explants remains not clear. 

Hydrogels are widely used in 3D tissue engineering as a promising strategy to restore damaged tissues. Various types of natural and synthetic compounds, or their combination-based hydrogels have been used for articular cartilage tissue regeneration [21]. Collagen II, as a most abundant component of the cartilage matrix, -based hydrogels were very popular in cartilage tissue engineering [22,23]. Electrospun gelatin, as a denatured form of collagen, has also been widely studied for cartilage reparative purposes [24]. It was shown that chondrocytes encapsulated into enzymatically crosslinked gelatin-hydroxyphenylpropionic acid (Gtn-HPA) conjugated the increased synthesis of GAGs and collagen and improved osteochondrial defect repair in rabbit model [25]. Another cartilage ECM component aggrecan is also made of sulfated glycosaminoglycans, including CS, which is responsible for high cartilage density [26]. Therefore, CS-based hydrogels can be among the most promising candidates to investigate the chondrogenesis of MSCs with or without mechanical load [27].

In addition, a persistent and strong mechanical stimulus can impair cartilage regeneration mechanisms through the dysregulation of intracellular calcium (iCa^2+^) ions applying various model systems in vitro and in vivo [28]. It is known that Ca^2+^ activated ion channels are essential for the physiological signaling and when dysregulated, can drive the impairment of cellular homeostasis and progress to pathophysiological sequalae [29]. Voltage-operated calcium channels (VOCCs) are responsible for the Ca^2+^ intake [30,31] and can be activated via mechanical load and other external stimuli affecting ECM, cell survival and differentiation-regulating signaling pathways [28]. On the other hand, the high level of cytoplasmic free iCa^2+^ is reduced by the ATPase pump-mediated iCa^2+^ sequestering into the endoplasmic reticulum (ER) [32]. Similarly, iCa^2+^ level regulating channels and/or their balance in chondrocytes and BMMSCs can be involved in cartilage regeneration mechanisms both in vitro and in vivo [33,34]. Moreover, it was shown, that a short-term increase in cytoplasmic free iCa^2+^ positively affects T cells and leads to synchronized iCa^2+^ waves, while prolonged and persistent elevation of iCa^2+^ results in cell apoptosis or other functional impairments [35]. However, how strong mechanical load affects iCa^2+^ regulating channels and the chondrogenic differentiation of BMMSCs and OA chondrocytes encapsulated into CS-based hydrogels and cartilage explants in vitro is not clear.

Therefore, the aim of this study was to compare chondrogenic responses of BMMSCs and OA cartilage-derived chondrocytes incorporated into CS-Tyr/Gel hydrogel under mechanical load in order to find out more about cartilage damaging mechanisms and possible targeted repairing means in vivo. 

## 2. Results

### 2.1. The Generation of CS-Tyr/Gel Hydrogel

The CS-Tyr/Gel-based hydrogel was generated by preparing two mixtures: (1) horseradish peroxidase (HRP) and 4% CS-Tyr, and (2) 5% gelatin and 3 mM H_2_O_2_. Then, the two solutions were mixed together. The gelation time of the mixed hydrogel components was measured by using an inversed vial test—the properly set hydrogel remained in an inverted tube.

The gelation time was longer when a higher concentration of H_2_O_2_ was used (Figure 1A), while a higher concentration of HRP resulted in a shorter gelation time (Figure 1B). The gelation test was conducted at 37 °C. If H_2_O_2_ concentration was lower than 3 mM at a stable 0.5 U/mL HRP concentration, the hydrogels did not set (Figure 1A). A higher HRP concentration resulted in a shorter gelation time (Figure 1B). As the HRP concentration was fixed at 0.3 U/mL, the increasing H_2_O_2_ concentration also obviously slowed down the gelation time. The data indicate that gelation time can be controlled by varying either the HRP or H_2_O_2_ concentrations. 0.3 U/mL of HRP and 3 mM of H_2_O_2_ were used for the following experiments.

### 2.2. The Characterization of Human BMMSCs and Chondrocytes

BMMSCs and chondrocytes were isolated from post-surgical biomaterial after obtaining the patients’ informed consent (the bioethics committee permission No 158200-14-741). BMMSCs had a typical spindle form of MSCs (Figure 2A) and surface biomarkers were positive for CD44, CD73, CD90 and CD105 (97 ± 2.5% of the total cell population) and negative for the hematopoietic stem cell markers CD14, CD34, CD36 and CD45 (less than 5 ± 2.5% of the total cell population). Chondrocytes were smaller in size (Figure 2A) and proliferated more slowly, as compared to BMMSCs (Figure 2B).

In addition, iCa^2+^ levels of BMMCSs and chondrocytes were identified by flow cytometer (Figure 3A), spectrophotometrically by measuring iCa^2+^ kinetics (Figure 3B) and microscopically (Figure 3C) by applying the fluorescent calcium-specific dye Cal-520. The level of cytoplasmic free iCa^2+^ was higher in chondrocytes than in BMMSCs, which could negatively affect proliferation of chondrocytes and increase their sensitivity to mechanical load. In addition, the more oriented growth of BMMSCs than OA chondrocytes may stabilize the cells.

### 2.3. The Encapsulation of BMMSCs and Chondrocytes into CS-Tyr/Gel Hydrogel and Viability Measurement

BMMSCs and chondrocytes were encapsulated into the CS-Tyr/Gel hydrogels by adding cells to the hydrogel mixture No. 1 (8% of CS-Tyr and 0.6 U of HRP) and mixed with the mixture No. 2 (4% of gelatin and 6 mM of H_2_O_2_) as written in Section 4. The cells and hydrogel formed round transparent drops that were easily observed under the light microscope (Figure 4A). The cells were evenly spread inside CS-Tyr/Gel hydrogel, as identified under the light microscope (Figure 4B).

The viability of encapsulated BMMSCs and chondrocytes was evaluated microscopically using a fluorometric Live/Dead kit (Figure 5A) and quantitatively by measuring metabolic cell activity as measured by cell counting kit 8 (CCK-8) kit (Figure 5B). Both types of encapsulated cells were viable (green cells) after 21 days of incubation in the chondrogenic differentiation medium (Figure 5A). A quantitative metabolic cell viability measurement confirmed the stable the encapsulated cell viability during 21 days of incubation in the chondrogenic differentiation medium (Figure 5B).

### 2.4. The Effect of Mechanical Load on Cell Viability and Release of Cartilage Oligomeric Matrix Protein (COMP) in Hydrogel/Cell Composites

Then, the viability and release of the ECM component COMP from BMMSCs and chondrocytes encapsulated into the CS-Tyr/Gel hydrogels was investigated with or without mechanical load (15 kPa, 1 Hz, 1 h/day for the last 7 days of differentiation), as described in Section 4. The increased release of LDH is related to plasma membrane damage and lower cell viability (Figure 6), which might account for the lower release of COMP (Figure 7) and the expression of chondrogenic genes *COL1A1* and *ACAN* as demonstrated in chondrocytes (Figure 8). Chondrocytes with or without TGF-β3 were more viable compared to the BMMSCs, but considerably more susceptible to mechanical load (Figure 6).

COMP is a glycoprotein, which assembles and stabilizes the extracellular matrix (ECM) via its interactions with type I and type II collagen and aggrecan. In the absence of TGF-β3, secretion of COMP in chondrocytes was stronger as compared to BMMSCs but was decreased after mechanical load (Figure 7). The stronger COMP secretion in cells is related to their better chondrogenic differentiation.

TGF-β3 significantly stimulated the release of COMP from both types of hydrogel/cell composites, indicating intensive chondrogenic differentiation, while mechanical load downregulated this effect (Figure 7). OA chondrocytes in hydrogels with or without TGF-β3, were more susceptible to mechanical load and secreted less COMP, as compared to BMMSCs (Figure 7).

Data suggest that the CS-Tyr/Gel/BMMSCs composite better suits the chondrogenic differentiation studies under mechanical load than the CS-Tyr/Gel/chondrocytes one. It also suggests that not all types of the cells and hydrogels can be used equally for cartilage regeneration purposes, particularly under mechanical load.

### 2.5. The Effect of Mechanical Load on Chondrogenic Differentiation Markers and iCa^2+^-Regulating Channels in Hydrogel/Cell Composites

Since the release of ECM component COMP decreased after mechanical load, it was interesting to investigate how the expression of other chondrogenic differentiation-related genes like collagen type II *(COL2A1*) and aggrecan *(ACAN)* changed subjecting to TGF-β3 and mechanical load. Data confirmed, that TGF-β3 stimulated expression of *COL2A1* and *ACAN* in both types of the hydrogel/cells composites (Figure 8A,B). However, CS-Tyr/Gel/chondrocytes with or without TGF-β3 were more sensitive to mechanical load compared to BMMSCs, as demonstrated by decreased expression of *COL2A1* and *ACAN* (Figure 8A,B). 

In addition, the expression of calcium L-type Ca^2+^ channel subunit CaV1.2 *(CACNA1C),* a regulator of Ca^2+^ entrance into the cell, and Serca2 pump *(ATP2A2),* a regulator of further iCa^2+^ accumulation into the ER, with or without TGF-β3 was stronger stimulated by mechanical load in BMMSCs than in chondrocytes (Figure 8C,D). In addition, the *CACNA1C/ATP2A2* ratio after 21 day of culture in chondrogenic conditions under different stimuli was more stable in CS-Tyr/Gel/BMMSCs than in CS-Tyr/Gel/chondrocytes revealing the tight iCa^2+^ control. Mechanical load downregulated the *CACNA1C/ATP2A2* ratio in chondrocytes, which might impair iCa^2+^ balance, leading to the altered chondrogenic phenotype (Figure 8E).

Both types of investigated cells can be used for the induction of chondrogenic differentiation in 3D CS-Tyr/Gel composites under regular conditions, while BMMSCs better survived mechanical load as compared to chondrocytes. 

### 2.6. The Effect of Mechanical Load on Chondrogenic Differentiation Markers and iCa^2+^-Regulating Channels in OA Cartilage Explants

In order to confirm the effect of mechanical load on CS-Tyr/Gel/BMMSCs and CS-Tyr/Gel/chondrocytes composites, we have investigated gene expression in human OA cartilage explants. The mechanical loading (30 kPa, 1 Hz, 1 h/day, for 7 days as described in Section 4) of OA cartilage explants significantly decreased expression of *COL2A1* and *ACAN* genes, while the expression of iCa^2+^ regulating channels *CACNA1C* and *ATP2A2* was significantly upregulated (Figure 9). The *CACNA1C/ATP2A2* ratio in OA cartilage explants after mechanical load was also impaired (from 0.03 ± 0.002 to 0.047 ± 0.003). These data suggest that strong mechanical load negatively affects the cartilage regeneration through changing the balance of iCa^2+^ regulating channels that could alter the level of cytoplasmic free iCa^2+^ and downregulate chondrogenic responses. The low expression level of *CACNA1C* compared to *ATP2A2* suggest the greater influence of the latter in the chondrogenesis.

## 3. Discussion

Articular cartilage is a specialized tissue, which absorbs external physiological or pathophysiological mechanical effects to further protect bone tissue [36]. Untreated cartilage injuries usually progress into degenerative joint diseases such as osteoarthritis and others [37]. OA cartilage defect repair has always been a vast problem due to its avascular nature and limited regenerative ability in situ. Therefore, various cartilage tissue engineering model systems in vitro have become a promising approach to investigate the regenerative potential of OA chondrocytes in combination with hydrogels, scaffolds, mechanical load and other components, opening new avenues for the prevention and repair of cartilage defects.

Cartilage is composed of chondrocytes and ECM, where chondrocytes are the only ones responsible for cartilage tissue integrity. From the category of differentiated cells with chondrogenic regenerative capacity, the mesodermal articular-derived chondrocytes (MDCs) and/or nasal crest-derived chondrocytes (NCDCs) are the most popular cartilage regenerating cells [38]. However, the nasal septum contains has few chondrocytes, which phenotype and response to mechanical load differs from the articular cartilage chondrocytes, as they originally function without any mechanical load. It was shown that the direct injection of fetal rabbits’ chondrocytes into rabbit OA knee cartilage improved its regeneration [39]. Chondroprogenitors (CPs) can be also isolated from various joint-related tissues such as meniscus, synovium or human epiphyseal, showing tissue specificity but limited cell number [40,41]. In addition, chondroprogenitors can be generated from induced pluripotent stem cells (iPSC) by editing *COL2A1* into the CRISP-Cas9 knock-in reporter cell line [42]. However, the multiple genetic modifications of hiPSC have limited their therapeutic application. As healthy adult human chondrocyte availability is limitted and culturing chondrocytes in monolayer (2D) changes their regenerative potential in vitro, 3D cell culturing for intraarticular injection systems are under investigation. Adults MSCs cells, particularly BMMSCs, have been extensively investigated for their cartilage regenerative applications [43]. However, similar to chondrocytes, the long-term 2D cultivation of adult MSCs were also prone to losing their cartilaginous phenotype [44]. Therefore, it increased the importance of developing new biomatrices mimicking 3D chondrogenic niche retaining chondrogenic cell phenotypes. In this study, we have compared chondrogenic differentiation and iCa^2+^ regulating mechanisms of BMMSCs and OA cartilage-derived chondrocytes, encapsulated into CS-Tyr/Gel hydrogel with or without mechanical load in vitro. In addition, the effects of mechanical load were also confirmed on human OA cartilage explants showing the similar molecular mechanisms damaging cartilage in vivo.

The hydrogels applied for the cartilage regeneration here are expected to have good injected cell survival/biocompatibility and resistance to mechanical load. However, various in vitro model systems of mechanical cell load exist, some of them mimic mild mechanical load, some are strong and might differently affect human cartilage. It was shown that BMMSCs encapsulated into collagen hydrogels and stimulated under static or dynamic conditions (a 10% peak compressive sinusoidal strain at 1 Hz frequency, for 2 h/day for 21 days) could activate the chondrogenic differentiation-related markers collagen II, aggrecan and Sox9 [45]. However, another study showed that dynamic long-term overload of synovial MSCs encapsulated into agarose hydrogel downregulated the expression level of chondrocyte-specific markers [46]. The efficiency of chondrogenic differentiation also strongly depends on the biocompatibility of the 3D cell-hydrogel composite during long term of incubation. So, many various factors influence mechanical load in vitro suggesting that there is no one universal compression system. If mechanical load worsens the functioning of the chosen experimental objects (hydrogels/cells, explants and other) evaluated at the ECM, intracellular protein and gene expression levels, it should be considered as a strong mechanical load or overload. And vice versa, the mild mechanical load usually stimulates synthesis of cartilage-specific proteins and genes. Data of this study show a good biocompatibility of CS-Tyr/Gel hydrogels with both BMMSCs and chondrocytes during 21 days of chondrogenic differentiation. Despite that, the OA chondrocytes embedded into CS-Tyr/Gel hydrogels were more sensitive to mechanical load (15 kPa, 1 Hz frequency 1 h/day for 7 days) compared to the BMMSCs, as indicated by the decreased secretion of COMP and lower expression of *COL2A1* and *ACAN* in those cells. Thus, CS-Tyr/Gel/BMMSCs seems more suitable for human cartilage regeneration under mechanical load conditions than CS-Tyr/Gel/chondrocytes. 

Having determined higher susceptibility of OA chondrocytes to mechanical load compared to the BMMSCs, we further explored the possible involvement of iCa^2+^ level-regulating L-type CaV1.2 and Serca2 channels. iCa^2+^ participates in various molecular mechanisms crucial for cell homeostasis, proliferation, differentiation, etc. [28,47]. In addition, Ca^2+^ is an important chondrogenesis regulating molecule, which enters cells mainly through the VOCCs and its subunit CaV1.2 [48,49]. Chondrocyte, similar to the excitable cells, also show spontaneous iCa^2+^ peaks in monolayer cultures or explants [50]. The causal relations between iCa^2+^ levels and ECM gene expression have been demonstrated by the suppression of mechanical loading-induced upregulation of many genes, including aggrecan, type II collagen, link proteins, c-Jun, and many MMPs by the presence of BAPTA-AM, a chelator of iCa^2+^ [51]. In addition, the expression of stress protein HSP70 was significantly upregulated, while expression of c-Jun was significantly suppressed during compression in the presence of BAPTA-AM. Furthermore, both mechanical and structural properties of the collagen fibril were shown to highly depend on the concentrations of calcium ions [52]. The mechanism of this dependence was attributed to the chelation between collagen molecules and calcium ions.

Since a high level of cytoplasmic free iCa^2+^ is usually cytotoxic to various types of cells [53], the higher iCa^2+^ level in chondrocytes, compared to BMMSCs, can be related to their higher sensitivity to mechanical load. On the other hand, the excess of iCa^2+^ is mainly stored in the ER, the entrance into which is regulated by the ATP-depended pump Serca2 [47]. Thus, iCa^2+^ is highly regulated by the proper balance between Ca^2+^ entrance into the cells and its accumulation into areas such as ER, that could regulate chondrogenic differentiation particularly under mechanical load [28]. It was shown that mechanical load and other extracellular stimuli can highly increase iCa^2+^ level by activating VOCC through the plasma membrane depolarization [54,55]. In addition, iCa^2+^ is necessary for the stabilization of COMP, also known as thrombospondin-5 (TSP-5), a soluble pentameric glycoprotein involved in the assembly and stabilization of the extracellular matrix via its interactions with type I and type II collagen [56]. So, the mechanical loading on articular cartilage can be transformed into many physical and chemical stimuli on chondrocytes residing in the ECM. ICa^2+^ signaling is among the earliest responses of chondrocytes to physical stimuli, but iCa^2+^ signaling in both differentiated cells or cartilage explants is not fully understood due to the technical challenges. Findings of this study suggest that mechanical load impaired the balance between iCa^2+^ regulating channels CaV1.2 and Serca2 in the CS-Tyr/Gel/chondrocytes composite and cartilage explants, which can be associated with the reduced release of COMP and lower expression of *COL2A1* and *ACAN.*

In summary, the data of this study show that in 2D control conditions, chondrocytes possessed a higher level of iCa^2+^, were smaller in size and had lower proliferation rate/metabolic activity than BMMSCs. In the 3D CS-Tyr/Gel hydrogel, chondrocytes secreted higher level of COMP, compared to the BMMSCs, which was reduced by mechanical load in both cell types, particularly strongly in chondrocytes. TGF-β3-stimulated chondrogenic gene expression in both hydrogel/cell composites, while mechanical load was more deleterious for the chondrogenic phenotype of CS-Tyr/Gel/chondrocytes, than of CS-Tyr/Gel/BMMSCs. Furthermore, mechanical load differently stimulated the expression of CaV1.2 and Serca2 encoding genes in the presence or absence of TGF-β3, resulting in altering their balance in CS-Tyr/Gel/chondrocytes and cartilage explants, modulation of iCa^2+^ levels and altered cartilage regenerating mechanisms. The CS-Tyr/Gel/BMMSCs composite seem to be more suitable for the cartilage regenerating purposes under mechanical load than CS-Tyr/Gel/OA cartilage-isolated chondrocyte.

## 4. Methods and Materials

### 4.1. CS-Tyr Synthesis

Tyramine (Tyr) was conjugated to chondroitin-4-sulfate sodium (CS) through the EDC/NHS (*N*-ethyl-*N*′-(3-(dimethylamino)propyl)carbodiimide/*N*-hydroxysuccinimide) reaction (Figure 10), according to the previously described method [57]. 

Briefly, 500 mg of chondroitin-4-sulfate sodium salt from bovine trachea (Sigma Aldrich, Burlington, MA, USA) was dissolved in 10 mL of MES buffer solution (19.52 g of 2-(*N*-morpholino)ethanesulfonic acid hydrate (MES) (Sigma Aldrich, Burlington, MA, USA) was dissolved in 1 L of deionized water and the pH was adjusted to 4.5, 100 mM), 141.9 mg of tyramine (Sigma Aldrich, Burlington, MA, USA) was dissolved in 20 mL of MES buffer solution (pH 4.5, 100 mM), 197.7 mg of 1-ethyl-3-(3-dimethylaminopropyl)carbodiimide (EDC) (Fluorochem, Hadfield, UK) and 119.2 mg *N*-hydroxysuccinimide (NHS) (Acros Organics, Morris Plains, NJ, USA) were dissolved in 10 mL of MES buffer solution (pH 4.5, 100 mM) and then added to the chondroitin-4-sulfate solution. After 30 min, all solutions were mixed and pH was adjusted to 4.5. The reaction mixture was stirred in the dark at room temperature (RT) for 24 h. The resulting solution was purified in dialysis membrane (MWCO 3500 Da) (Spectra/Por^®^ 7 MWCO 3500, 45 mm (Carl-Roth, Karlsruhe, Germany) against acidified deionized water (pH ~ 3) four times under gentle stirring and finally against deionized water (pH ~ 7) for 2 h under vigorous stirring. The final product was lyophilized and stored at −20 °C. The degree of substitution of tyramine in chondroitin sulfate was assessed via the absorbance of phenolic groups at 276 nm.

### 4.2. Formation and Investigation of CS-Tyr/Gel Hydrogel

The formation of the CS-Tyr/Gel hydrogel was mediated by an enzyme-induced oxidative coupling reaction as written in [57], with some modifications, which connects the phenolic functionality between the CS-Tyr and gelatin. Briefly, 8% of CS-Tyr and 4% of gelatin (Extra Pure, SLR (Fisher Chemical, Waltham, MA, USA) were separately dissolved in 10× PBS at RT, and pH was adjusted to 7.4. Then two mixtures were prepared: (1) the 8% of CS-Tyr and 0.6 U/mL of HRP (Sigma Aldrich, Burlington, MA, USA); (2) the 4% of gelatin and 6 mM of hydrogen peroxide solution ≥ 30% (H_2_O_2_) (Sigma Aldrich, Burlington, MA, USA). The prepared two mixtures were mixed in a 1:1 ratio and allowed to stay at RT for 10 min. The gelation time of the mixed hydrogel components was measured using the inversed vial test—the properly set hydrogel remained in the vial.

The hydrogel consisting of CS-Tyr and various amounts of gelatin and their elasticity were analyzed using dynamic mechanical analysis (Modular compact rheometer, MCR-102, Anton Paar, QC, Canada). The mechanical test was operated in a frequency sweep mode of 0.1–10 Hz at 1% strain and 37 °C. The elasticity of the CS-Tyr/gelatin hydrogels was not affected with an addition of small amount of gelatin. When the gelatin content increased to 1%, the elastic modulus decreased about 50% (Figure 11). However, the elastic moduli did not further decrease as the gelatin content increased to 4%. The decrease in elastic moduli might be due to a low tyrosine content of gelatin. Since gelatin supports cell activity, 4% gelatin was used in the subsequent cell experiments. The swelling ratio of the CS-Tyr/gelatin hydrogel based on its dried weight was about 30~35-fold.

For BMMSCs incorporation into CS-Tyr/Gel hydrogel, 250 thousand of the harvested cells were washed with phosphate-buffered saline (PBS) (Sigma Aldrich, Burlington, MA, USA), centrifuged and suspended into the hydrogel mixture No 1. Then the two mixtures were mixed in a 1:1 ratio and polymerized at RT for 10 min. The polymerized CS-Tyr/Gel composites were transferred to the 6 well Flexcell plate with a regular cell growth medium and incubated at 37 °C and 5% CO_2_ for 14 days. The medium was changed twice a week. Then the hydrogels were subjected to mechanical load (15 kPa, 1 Hz, 1 h/day) for the next 7 days and further investigated with various assays.

### 4.3. Cell Isolation and Culture

BMMSCs were isolated from healthy human bone marrow samples (*n* = 5), which were received from Vilnius University Hospital Santaros Klinikos after joint surgery. Bone marrow samples were washed with PBS (Sigma Aldrich, Burlington, MA, USA) containing 1% of penicillin-streptomycin (PS) (Gibco, Life Technologies, Waltham, MA, USA) solution, transferred to a sterile dish, and one more time washed with PBS/PS. Bone marrow was excluded from the bone and placed in low glucose (1 g/L) Dulbecco’s modified Eagle’s medium (DMEM) (Capricorn Scientific, Ebsdorfergrund Germany) with 1% PS. Afterwards, the bone marrows were chopped to a liquid consistency and mixed in DMEM/PS medium. The obtained suspension was filtered through 100 µm filter and centrifuged for 10 min at 350× *g*. The cell pellet was resuspended in complete growth medium, consisting of DMEM (1 g/L glucose) supplemented with 10% FBS (Gibco, Life Technologies, Waltham, MA, USA) and 1% PS, additionally supplemented with fibroblast grow factor 2 (FGF-2) (20 ng/mL) (Sigma Aldrich, Burlington, MA, USA), counted and cultured at regular cell cultivating conditions—37 °C and 5% CO_2_.

Chondrocytes were isolated from post-operative human articular cartilage samples (*n* = 5), received from Vilnius University Hospital Santaros Klinikos after OA patients’ joint surgery. Tissue samples were cut off from the most healthy areas of cartilage. Cartilage samples were washed with PBS containing 1% of penicillin-streptomycin (PS) solution and chopped into small pieces, with an average size of 1 mm^2^. Minced cartilage tissue was incubated in DMEM (1 g/L glucose) with 1% PS at 37 °C and 5% CO_2_ overnight. After the incubation, cartilage tissue was washed with PBS. Cells from cartilage tissue samples were isolated enzymatically. Firstly, cartilage tissue was incubated in pronase (Sigma Aldrich, Burlington, MA, USA) solution for 1 h at 37 °C and 5% CO_2_. After the incubation, the digested tissue was washed twice with PBS and chopped to homogeneity. The mass of cartilage tissue was transferred into a 50 mL tube and digested with a type II collagenase solution (545 U/mL) (Biochrom AG, Berlin, Germany) 10 milliliters/1 g of the cartilage sample. Chondrocyte isolation was performed for 4 h at 37 °C and 5% CO_2_, under constant shaking. After the isolation of chondrocytes, 100 and 70 µm strainers were used to remove undigested cartilage tissue. The cell suspension was centrifuged at 400× *g* for 5 min and the cell pellet was suspended in complete growth medium used for BMMSCs, just without FGF-2. Isolated chondrocytes and BMMSCs were cultured in flasks (Orange Scientific, Braine-l’Alleud, Belgium) with a complete medium in a 37 °C incubator with 5% CO_2_, changing the medium twice a week.

### 4.4. The Viability and Metabolic Activity of Cells Encapsulated into CS-Tyr Hydrogel

Cell viability in hydrogels was evaluated by using the commercial Live/Dead cell viability assay kit (Thermo Fischer Scientific, Waltham, MA, USA). The kit contains fluorescent Calcein-AM dye, which reacts with an intracellular esterase and live cells become fluorescent green, while ethidium homodimer-1 interposes into the DNA of dead cells and those cells become fluorescent red.

The CS-Tyr hydrogels with incorporated cells (volume of hydrogel 50 µL, cell concentration 1 × 10^6^ cells/mL cells/mL) were placed into 96 well plates. Set hydrogels were cultured with 100 µL of complete medium in a 37 °C incubator with 5% CO_2_. The medium was changed every two days. After 1, 3, and 7 days, the hydrogels were incubated with Live/Dead solution for 15 min. After incubation, cell viability was evaluated with the fluorescent microscope EVOS M7000 (Waltham, MA, USA).

Additionally, the cell proliferation/metabolic cell activity in hydrogels was determined at 1, 7, 14, and 21 days using commercially available CCK-8 (Dojindo, Munich, Germany) according to the manufacturer’s instructions. CCK-8 kit allows us to measure cell proliferation and/or cytotoxicity at once by utilizing water-soluble reduced tetrazolium salt. This salt is reduced by intracellular dehydrogenases and produces an orange-colored formazan dye measured spectrophotometrically (SpectraMax i3, Molecular Devices, San Jose, CA, USA) at 450 nm. The intensity of absorption of formazan dye generated by cell dehydrogenases was directly proportional to the number of living cells. Three technical replicates of three donor cells of each type were measured.

### 4.5. Analysis of Intracellular Calcium Level

For the iCa^2+^ analysis with the fluorescent microscope, the cells were seeded into 24-well plates at a density of 10,000 cells/well. After the cells reached confluence, they were stained with calcium specific fluorescent dye Cal-520 (1 μM) (Santa Cruz, Biotechnologies, Dallas, TX, USA) in Hanks’ buffered saline solution (HBSS) with probenecid (1 mM) (Sigma Aldrich, Burlington, MA, USA) and Pluronic^®^ F-127 (0.02%) (Sigma Aldrich, Burlington, MA, USA) and incubated at 37 °C with 5% CO_2_ for 90 min. After this, the cells were washed twice with PBS and HBSS with 1 mM probenecid (Sigma Aldrich, Burlington, MA, USA) added before cell analysis with a fluorescent microscope (EVOS M7000, Waltham, MA, USA).

In addition, iCa^2+^ was measured spectrophotometrically (SpectraMax i3, Molecular Devices, San Jose, CA, USA) for 10 min to evaluate the kinetics of iCa^2+^ change in the cells (E_x_/E_m_ = 490/525 nm, background subtracted at 488 nm) using the same Cal-520 (1 μM).

For the iCa^2+^ analysis with a flow cytometer, the cells were seeded into 6-well plates at a density of 50,000 cells/well. After the cells reached confluence, they were detached with trypsin/0.25% EDTA (Sigma Aldrich, Burlington, MA, USA) and 50,000 of cells were transferred into flow cytometry tubes, centrifuged at 500× *g* for 5 min and stained with the same calcium specific fluorescent dye Cal-520 (1 μM) (Santa Cruz, Biotechnologies) in DMEM medium with 10% of FBS at 37 °C with 5% CO_2_ for 30 min. After incubation, the cells were washed with PBS and centrifuged at 500× *g* for 5 min. The supernatant was discarded and cells were resuspended in 300 μL of PBS with 1% of bovine serum albumin (BSA) (Sigma Aldrich, Burlington, MA, USA). Each sample was run in triplicates. The fluorescence was measured using a flow cytometer (FACSAria III, Franklin Lakes, NJ, USA) and evaluated by Diva software (v9.0, Franklin Lakes, NJ, USA).

### 4.6. Chondrogenic Differentiation of Cells Encapsulated into CS-Tyr/Gel with/without the Mechanical Loading

Chondrogenesis was induced using a protocol developed at the State Research Institute Centre for Innovative Medicine, Lithuania [4]. Briefly, the chondrogenic medium was composed of high glucose (4.5 g/L) DMEM medium, 1% PS, 1% insulin-transferrin-selenium (Gibco, Life Technologies, Waltham, MA, USA), 350 μM L-proline (Carl Roth, Karlsruhe, Germany), 0.1% dexamethasone, 170 μM ascorbic acid-phosphate (Sigma Aldrich, Burlington, MA, USA) and 10 ng/mL TGF-β3 (Gibco, Life Technologies, Waltham, MA, USA). BMMSCs or chondrocytes were incorporated into the CS-Tyr/Gel hydrogels and transferred into two different 6 well plates (50 × 10^3^ of the cells in 50 μL of CS-Tyr for one well). One plate was used for the mechanical compression with the commercial Flexcell system plate, while the another was used as control.

The chondrogenic medium was added on the same day (3 mL/well). The plates with hydrogels/cells were divided into two groups (3 wells/group): (1) Control (hydrogels/cells with chondrogenic medium without TGF-β3); (2) Differentiated (hydrogels/cells with chondrogenic medium and with growth factor TGF-β3). The medium was changed three times a week. Two weeks after chondrogenic differentiation started, the Flexcell plate with hydrogels/cells was transferred into the “Flexcell FX-5000” (Burlington, NJ, USA) system and was compressed using a square shaped dynamic function with 15 kPa, 1 Hz compression, at 1 h/day for 7 days. The standard control plates were not mechanically compressed. After 7 days of stimulation, Flexcell and standard plates with hydrogels/cells were analyzed for the secreted proteins and intracellular gene expression. The ECM in the hydrogels/cells biomatrices was evaluated using ELISA assay for detection of the secreted COMP (Biovendor, Brno, Czech Republic) and chondrogenic gene expression measurement. 

The viability of the cells encapsulated into hydrogel for 21 days with or without the chondrogenic medium and compression was estimated spectrophotometrically by the released amount of lactate dehydrogenase (LDH) (Thermo Fischer Scientific, Waltham, MA, USA) according to the manufacturer’s recommendations. The release of LDH is related to the level of plasma membrane damage, i.e., a lower LDH release represents better cell viability.

### 4.7. Isolation and Mechanical Compression of Human Cartilage Explants

Fragments of human OA articular cartilage were obtained from patients undergoing joint replacement surgery at Vilnius Santaros Hospital and transported in a sterile container with PBS (Sigma Aldrich, Burlington, MA, USA) (Bioethics committee permission No. 158200-14-741). All further work was carried out in the laboratory laminar under sterile conditions. The cartilage was washed several times with PBS containing 2% PS and transferred to a Petri dish. Using a biopsy needle, the cartilage was cut into flat, round explants of 3 mm diameter and 3 mm height and each explant was weighed. The cartilage explants were washed and transferred into two wells of 6 well plates (120 mg of cartilage into one well), i.e., one well for mechanical load of “Flexcell FX-5000” (Burlington, NJ, USA) system, while another one was a control plate. Cartilage explants were cultured in chondrogenic medium.

The next day, the plate with cartilage explants was transferred to the mechanical compression system Flexcell FX-5000 placed in the incubator with 5% CO_2_ at 37 °C and compressed using a square-shaped dynamic compression mode with 30 kPa and 1 Hz frequency strength, at 1 h/day for 7 days. The explants’ growth medium was changed twice a week. After 7 days, the explants from each study group with or without mechanical load were taken for further PCR, COMP, GAG and LDH assays.

Chondrocyte viability in explants was analyzed using the released amount of lactate dehydrogenase (LDH) (Thermo Fischer Scientific, Waltham, MA, USA) according to the manufacturer’s recommendations, using the spectrophotometer SpectraMax i3 (Molecular Devices, San Jose, CA, USA). LDH is directly related to the damage of the cells’ plasma membrane, i.e., a lower LDH release represents better cell viability.

### 4.8. Secretion of Cartilage Oligomeric Matrix Protein (COMP)

The secreted COMP was evaluated 21 days after the exposure of the encapsulated BMMSCs and chondrocytes to chondrogenic differentiation medium with or without TGF-β3 and mechanical load. Supernatants (3 days after the last medium change) were collected and the levels of COMP were estimated using COMP ELISA (Biovendor, Brno Czech Republic) according to the manufacturer’s instructions. The absorbance was measured at 450 nm using the spectrophotometer SpectraMax i3 (Molecular Devices, San Jose, CA, USA).

### 4.9. RNA Extraction from the CS-Tyr/Gel/Cells and Cartilage Explants

After chondrogenic differentiation, the hydrogels/cells and cartilage explants were collected, flash-frozen in liquid nitrogen and stored at −70 °C. Frozen samples were homogenized using an ultrasonication system (Bandelin Sonopuls, Burlington, MA, USA) in lysis buffer (Qiagen, Venlo, The Netherlands) and RNA was extracted according to the manufacturer’s protocol. The RNA concentration and purity of all samples was measured with a SpectraMax i3 (Molecular Devices, San Jose, CA, USA).

### 4.10. RT-qPCR

RNA was reverse-transcribed with a Maxima cDNA synthesis kit including dsDNase treatment (Thermo Fischer Scientific, Waltham, MA, USA). RT-qPCR reaction mixes were prepared with the Maxima Probe qPCR Master Mix (Thermo Fischer Scientific, Waltham, MA, USA) and TaqMan Gene expression Assays (*RPS9*—Hs02339424_g1, *B2M*—Hs00984230_m1, *COL2A1*—Hs01060345_m1, *ACAN*—Hs00153936_m1, *CACNA1C*—Hs00167681_m1, *ATP2A2*—Hs00544877_m1 (Thermo Fischer Scientific, Waltham, MA, USA), and run on the Agilent Aria MX instrument (Agilent Technologies, Santa Clara, CA, USA) in technical triplicates starting with denaturation step at 95 °C for 10 min followed by 40 cycles at 95 °C for 15 s of denaturation and 60 s for annealing and extension. Relative levels of gene transcripts were calculated by subtracting the threshold cycle (Ct) of the normalizer (the geometric mean of the two housekeeping genes—RPS9 and B2M) from the Ct of the gene of interest, giving dCt values that were subsequently transformed to 2^−dCt^ values and multiplied by 1000 to scale-up for better graphical representation.

### 4.11. Statistical Analysis

The results are presented by the mean ± standard deviation (SD) from three repeats of not less than three cell cultures. Data are significant at *p* value of ≤0.05 calculated using Excel (16.69) and Prism (8.4.0) softwares.

## 5. Conclusions

The CS-Tyr/Gel hydrogel seems to be a suitable hydrogel for the induction of the chondrogenic responses of both encapsulated BMMSCs and chondrocytes under stimulation with TGF-β3, as determined by the increased secretion of COMP and expression of collagen type II (*COL2A1*) and aggrecan (*ACAN*). The CS-Tyr/Gel/chondrocytes under 3D chondrogenic conditions in the CS-Tyr/Gel with or without TGF-β3 more intensively expressed collagen type II (*COL2A1*) and aggrecan (*ACAN*) compared to the CS-Tyr/Gel/BMMSCs but were more susceptible to mechanical load, as determined by the reduced cell viability, COMP secretion, lower expression of collagen type II and aggrecan. In BMMSCs, the expression of *COL2A1* was suppressed by mechanical load in the absence of TGF-β3, while upregulated in the presence of TGF-β3.

In addition, under mechanical load, iCa^2+^ in CS-Tyr/Gel/BMMSCs seems to be more stably controlled through the expression of L-type channel subunit CaV1.2 (*CACNA1C*) and Serca2 (*ATP2A2*) pumps, as well as maintaining their balance composites more than in CS-Tyr/Gel/chondrocytes one. Our data suggest that BMMSCs are less susceptible to mechanical load than chondrocytes in CS-Tyr/Gel hydrogel, and therefore may have advantages in application for cartilage regeneration purposes. The cartilage damage due to the mechanical overload of OA chondrocytes in vivo and vague regenerative processes might be related to the less efficient control of iCa^2+^-regulating channels.

Apparently, not all types of the cells and hydrogels can be equally used for cartilage regeneration purposes, particularly under mechanical load. The tight balance of iCa^2+^-regulating channels is involved in promotion of cartilage regeneration under mechanical load. Mechanical load on OA cartilage explants and/or chondrocytes should be used with caution.

## Figures and Tables

**Figure 1 ijms-24-02915-f001:**
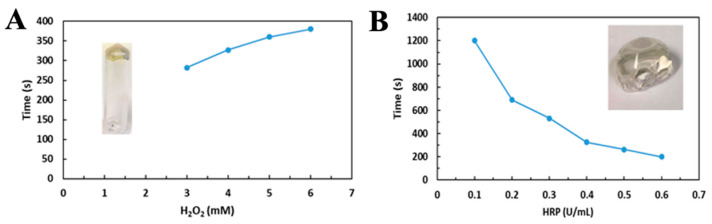
The investigation of gelation conditions of CS-Tyr/gelatin hydrogel. (**A**) Gelation time of 0.5 U HRP/mL and various H_2_O_2_ concentrations. (**B**) Gelation time of 3 mM H_2_O_2_ and various HRP concentrations.

**Figure 2 ijms-24-02915-f002:**
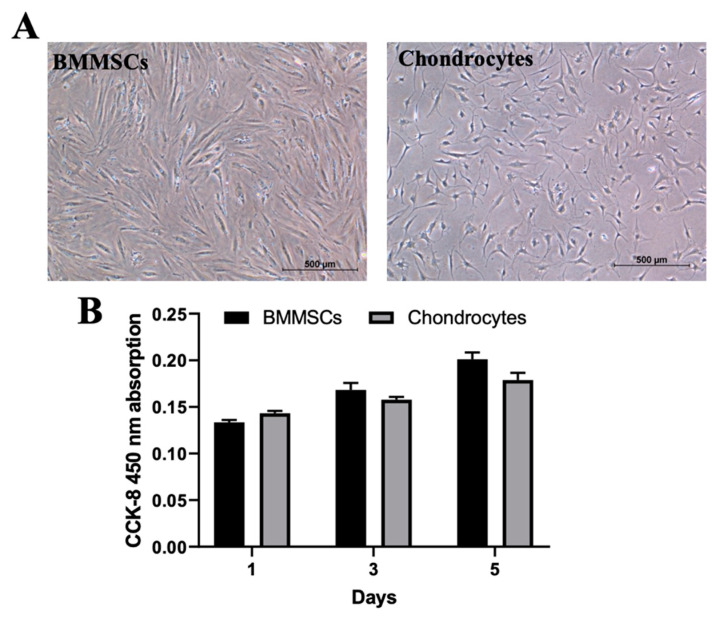
Characterization of BMMSCs and chondrocytes. (**A**) Light microscope micrographs of BMMSCs and chondrocytes, magnification: ×40; (**B**) Proliferation of BMMSCs and chondrocytes using CCK-8 kit and measuring absorption at 450 nm.

**Figure 3 ijms-24-02915-f003:**
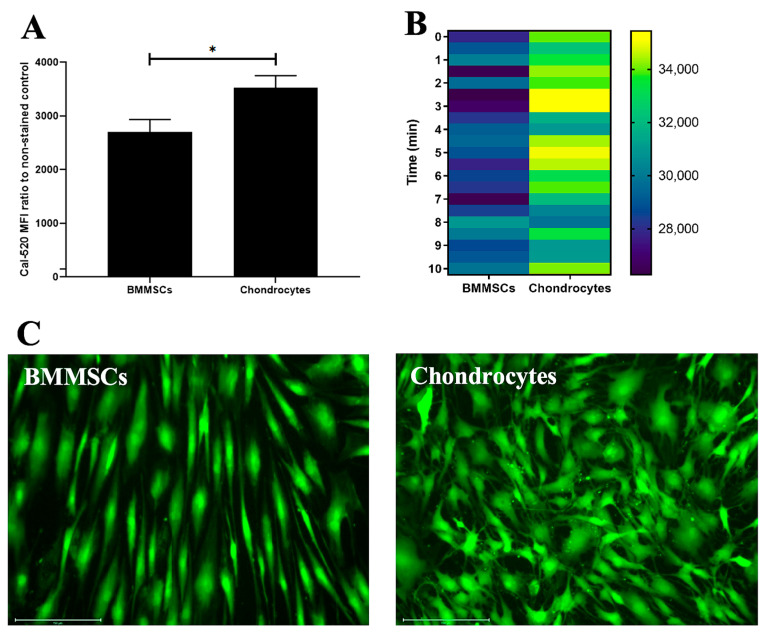
iCa^2+^ in BMMSCs and chondrocytes stained with Cal-520. (**A**) Flow cytometry analysis showing medial fluoresce intensity (MFI) as ratio to non-stained cells (BD FACSAria). Data are shown as mean ± SD, *n* = 3 from not less than three different cultures of each type and are significant at * *p* ≤ 0.05. (**B**) iCa^2+^ kinetics of BMMSCs and chondrocytes cultivated in a 2D monolayer, stained with Cal-520 and measured with a spectrophotometer (SpectraMax i3, Molecular Devices, San Jose, CA, USA). (**C**) The micrographs of BMMSCs and chondrocytes cultivated in a 2D monolayer, stained with Cal-520 dye and measured with an EVOS fluorescent microscope (Waltham, MA, USA). Scale bar 200 μm.

**Figure 4 ijms-24-02915-f004:**
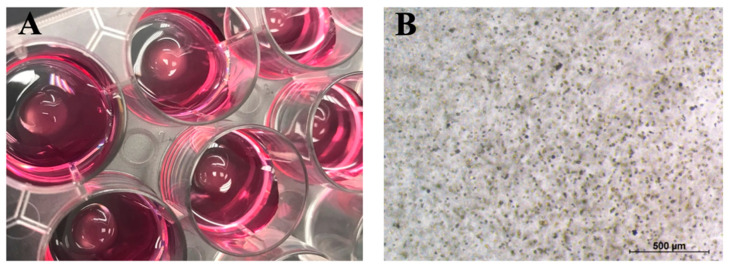
The generation of CS-Tyr-Gel/Cells composites. (**A**) Set CS-Tyr/Gel hydrogel with the encapsulated cells. (**B**) Visualization of cell distribution inside the hydrogel composite under the light microscope. The representative CS-Tyr/Gel/BMMSCs composite is shown.

**Figure 5 ijms-24-02915-f005:**
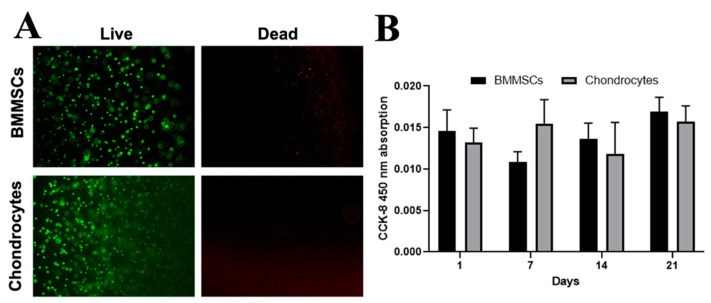
The viability of BMMSCs and chondrocytes in CS-Tyr/Gel hydrogels subjected to the chondrogenic differentiation for 1, 7 and 14 days. (**A**) The fluorescent micrographs of the encapsulated cells using Live/Dead kit: Calcein-AM shows viable cells (green fluorescence), while propidium iodide identifies dead cells (red fluorescence) after 21 days in chondrogenic differentiation medium; magnification: ×40. (**B**) The metabolic cell activity measurement using commercial CCK-8 kit (cell medium absorption was spectrophotometrically measured at 450 nm). Data are shown as mean ± SD, *n* = 3 from not less than three different cell cultures of each type.

**Figure 6 ijms-24-02915-f006:**
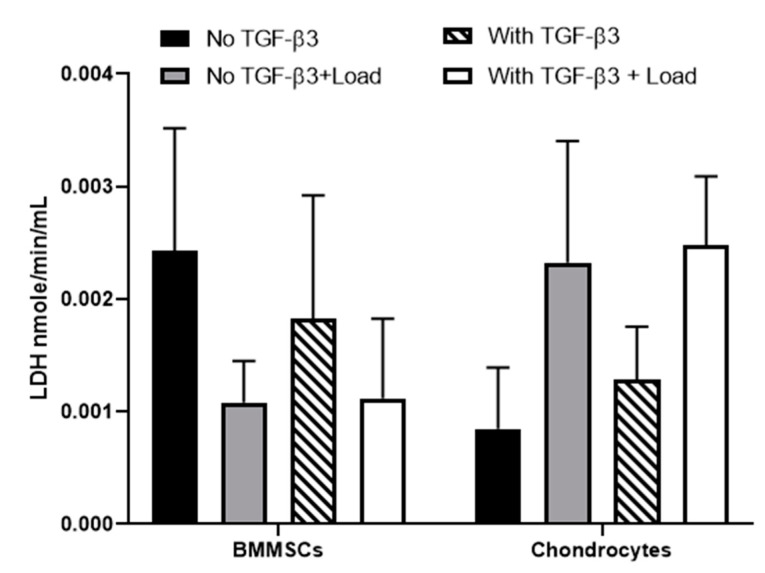
The viability of BMMSCs and chondrocytes encapsulated into the CS-Tyr/Gel hydrogel. The level of secreted lactate dehydrogenase (LDH) measured using ELISA, after the chondrogenic differentiation for 21 days with/without mechanical load for 7 days. The cells were cultivated in chondrogenic medium with or without TGFβ3 and mechanical load. Data are shown as mean ± SD, *n* = 3 from not less than three different cell cultures of each type.

**Figure 7 ijms-24-02915-f007:**
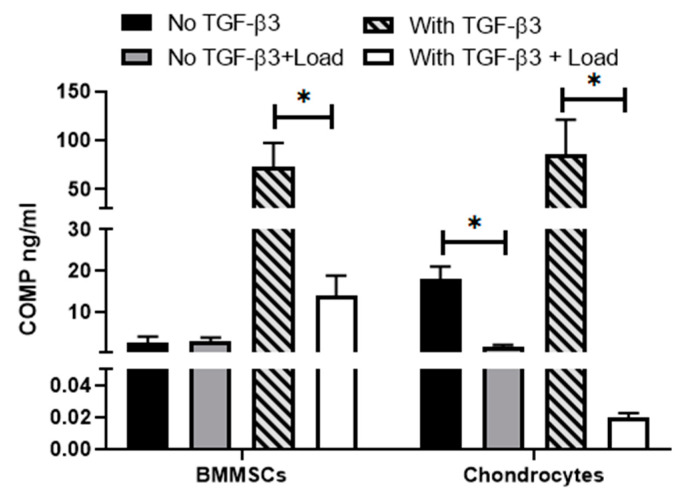
The release of oligomeric matrix protein (COMP) by the BMMSCs and chondrocytes encapsulated into the CS-Tyr/Gel hydrogel. The levels of secreted COMP after the chondrogenic differentiation for 21 days with/without mechanical load. Data are shown as mean ± SD, *n* = 3 from not less than three different cultures of each type and are significant at * *p* ≤ 0.05. The cells were cultivated in chondrogenic medium with or without TGFβ3 and mechanical load.

**Figure 8 ijms-24-02915-f008:**
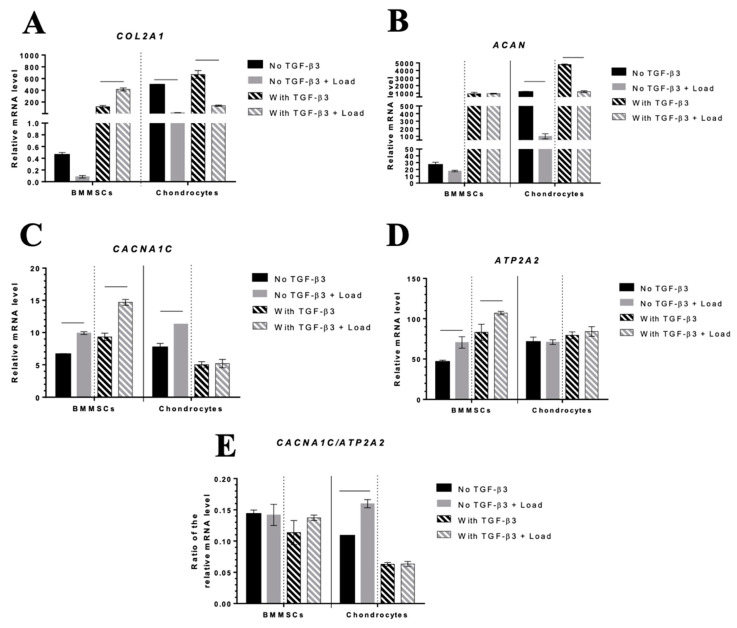
The effect of TGF-β3 and mechanical load on CS-Tyr/Gel/BMMSCs and CS-Tyr/Gel/chondrocytes composites. The expression of chondrogenic differentiation-related markers, (**A**) collagen II *(COL2A1)*, (**B**) aggrecan *(ACAN)*, and iCa^2+^ regulators, (**C**) L-type channel subunit CaV1.2 *(CACNA1C)* and (**D**) Serca2 pump *(ATP2A2),* were analyzed on 21st day of chondrogenic differentiation with/without TGF-β3 (10 ng/mL) and mechanical load, as described in the Methods section. (**E**) Ratio of *CACNA1C/ATP2A2*. Data are shown as mean ± SD from three repeats of two BMMSC and chondrocyte cell cultures and are significant at *p* ≤ 0.05 (horizontal bars). The cells were cultivated in chondrogenic medium with or without TGF-β3 and mechanical load. Gene expression was normalized with B2M and RPS9 housekeeping genes.

**Figure 9 ijms-24-02915-f009:**
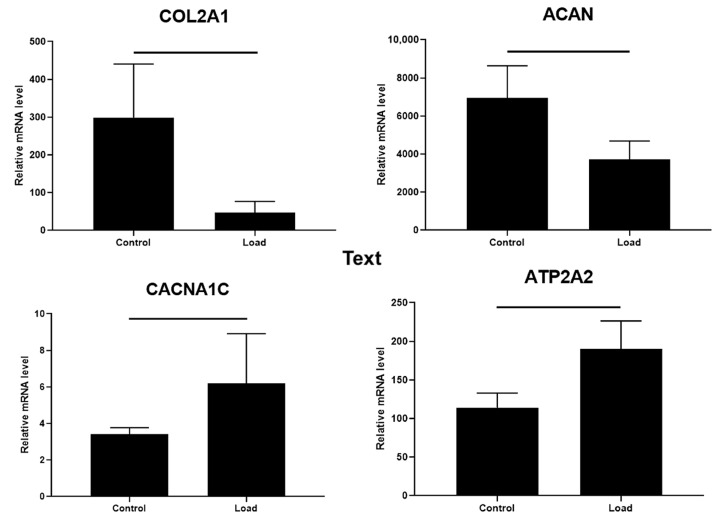
The effect of mechanical load on human OA cartilage explants. The expression of collagen 2 *(COL2A1),* aggrecan *(ACAN),* L-type channel subunit CaV1.2 *(CACNA1C)* and Serca2 pump *(ATP2A2)* genes in cartilage explants under regular growth conditions with/without mechanical load as described in the Methods section. Relative transcript level after normalization to the geometric mean of housekeeping B2M and RPS9 genes are shown. Data are presented as mean ±  SD and are significant at *p* ≤ 0,05 from not less than three repeats of three patients’ cartilage explants. Control—cartilage samples without load.

**Figure 10 ijms-24-02915-f010:**
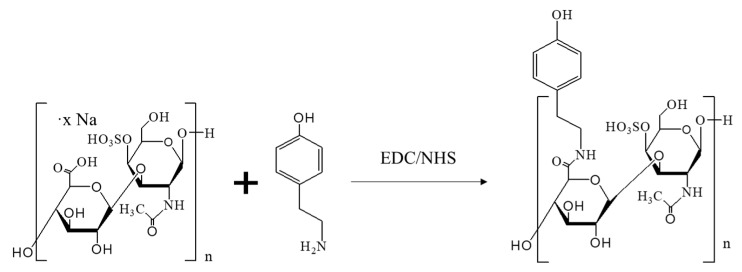
Chemical synthesis of CS-Tyr through EDC/NHS reaction.

**Figure 11 ijms-24-02915-f011:**
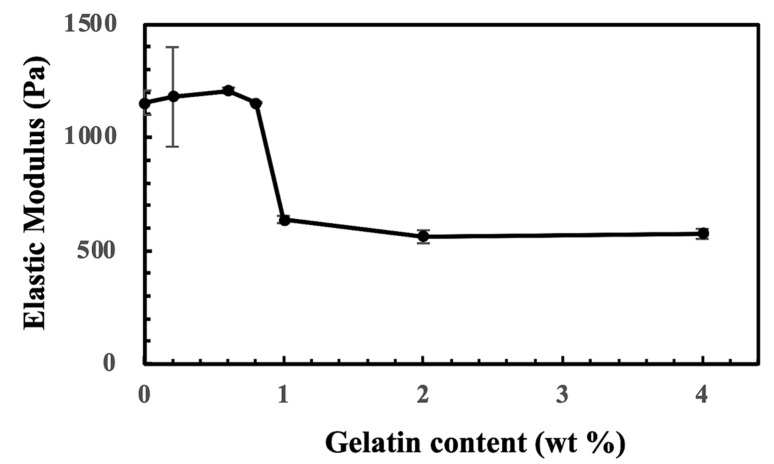
The elastic moduli of CS-Tyr/gelatin hydrogels, containing 4% CS-Tyr and 0–4% of gelatin. *n* = 3, error bars represent standard deviations.

## Data Availability

The data supporting these findings can be found at State Research Institute Centre for Innovative Medicine, Department of Regenerative Medicine.

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
