# Peer review of "The Effects of Mechanical Load on Chondrogenic Responses of Bone Marrow Mesenchymal Stem Cells and Chondrocytes Encapsulated in Chondroitin Sulfate-Based Hydrogel"

_ijms, 2023, doi:10.3390/ijms24032915_

Round 1

Reviewer 1 Report

The current manuscript compares the chondrogenic effects of hydrogel encapsulated OA chondrocytes and BMMSCs under mechanical overload. The possible mechanisms are also investigated. The overall study is interesting and may invoke some deeper thinking regarding the relationship between mechanical stimulation and cartilage regeneration.  

General comment:

Compared to OA chondrocytes, BMMSCs are much easier to obtain and have much higher proliferation ability. Therefore, readers are very easy to reach the conclusion that: compared with OA derived chondrocytes, BMMSCs are more suitable as seed cells to treat OA through a tissue engineering approach. So, what exactly is the point in this study to still compare these two types of cells? Im more curious whether healthy normal chondrocytes derived from hyaline cartilage, such as chondrocytes from healthy articular joint or even from the nasal septum, would be more tolerant to the mechanical load than the OA derived chondrocytes. Also, what would these cells (healthy chondrocytes, OA chondrocytes and BMMSCs) behave under mild mechanical load as well as under strong overload?

Specific comment:

1. The authors may want to provide more information on the mechanical load used in this study. Why the mechanical load used in this study be determined as overload? what is the corresponding mild mechanical load? Again, what would the cells behave under the mild load and the strong overload?

2. Figure 6,7: the colors representing “No TGF-β3+Load” and “With TGF-β3+Load” are too similar with each other, making it a little difficult to discriminate.  

3. What exactly is the relationship between iCa2+ and the down regulation of cartilage specific matrix? Is it really a causal relationship? Is it possible that another  reason that truly caused the simultaneous up regulation of iCa2+ and down regulation of the cartilage specific matrix?

Author Response

We thank the reviewer for the valuable comments and suggestions. Please find attached our answers to each question and a manuscript updated accordingly. We hope the reviewer will accept our answers and corrections.

Reviewer 1:

The current manuscript compares the chondrogenic effects of hydrogel encapsulated OA chondrocytes and BMMSCs under mechanical overload. The possible mechanisms are also investigated. The overall study is interesting and may invoke some deeper thinking regarding the relationship between mechanical stimulation and cartilage regeneration.

General comment:

Compared to OA chondrocytes, BMMSCs are much easier to obtain and have much higher proliferation ability. Therefore, readers are very easy to reach the conclusion that: compared with OA derived chondrocytes, BMMSCs are more suitable as seed cells to treat OA through a tissue engineering approach. So, what exactly is the point in this study to still compare these two types of cells? I’m more curious whether healthy normal chondrocytes derived from hyaline cartilage, such as chondrocytes from healthy articular joint or even from the nasal septum, would be more tolerant to the mechanical load than the OA derived chondrocytes. Also, what would these cells (healthy chondrocytes, OA chondrocytes and BMMSCs) behave under mild mechanical load as well as under strong overload?

Answer: The OA chondrocytes, used in this study, can be easier obtained from the patients’ OA cartilage, due to the routine surgeries in hospitals. To obtain healthy cartilage chondrocytes is complicated, as surgeries of healthy joints are performed only in rare cases of trauma. Similarly, the obtaining of healthy human bone marrow can be also complicated and number of BMMSCs in received samples is usually low. This is why human autologous/allogeneic chondrocytes are often used in the pre-clinical/clinical studies for the cartilage tissue regeneration purposes.

            However, the cartilage repair potential of BMMSCs (since they are obtained from the healthy human) are more promising due to their better ability to re-build cartilage ECM after the differentiation into chondrocytes, compared to the other types of adult MSCs or OA chondrocytes. On the other hand, the nasal septum also contains small amount of cells, which phenotype and response to the mechanical load differs from the articular cartilage chondrocytes, as they originally function without any mechanical load. Therefore, comparing the nasal septum-derived chondrocytes with the articular chondrocytes under the mechanical load would not be completely scientifically correct.

OA chondrocytes, in this study, were used as a native cartilage cell control for the BMMSCs encapsulated into the CS-Tyr/Gel hydrogel with/without the mechanical load. The response of OA chondrocytes to the mechanical load was worse as compared to the encapsulated BMMSCs – it is probably self-understandable since bot tissues are different in terms of patients’ health. However, data of this study show that at control level in the OA chondrocytes in 3D CS-Tyr/Gel hydrogel secreted higher levels of COMP, as compared to the BMMSCs, however, under the mechanical load, the COMP level in OA chondrocytes was stronger reduced than in BMMSCs. Data suggest that OA chondrocytes originally have higher chondrogenic potential than BMMSCs, but are more sensitive to the mechanical load.

We assume that the response of healthy chondrocytes to the mechanical load would be similar to the differentiated healthy BMMSCs. We also assume, that healthy cartilage should be less susceptible to the mechanical load compared to the OA cartilage, as well as encapsulated healthy BMMSCs less susceptible compared to the OA cartilage-derived chondrocytes. So, the healthy tissues or derived cells always better function than the pathological tissues or cells.

Therefore, in this study we wanted to compare the response of two types of cells encapsulated in the CS-Tyr/Gel hydrogels to the mechanical load and evaluate the changes of their iCa2+ regulating systems in comparison to the response of native cartilage explants to the mechanical load. This type of study so far was not done.

Specific comment:

  1. The authors may want to provide more information on the mechanical load used in this study. Why the mechanical load used in this study be determined as “overload”? what is the corresponding mild mechanical load? Again, what would the cells behave under the mild load and the strong overload?

Answer: Thank you for the question. There are various types of mechanical compression systems in vitro mimicking loads that cartilage suffers during human walk or intensive sport. The mechanical compression used in this study negatively affected both the encapsulated cells and OA cartilage explants, and could be named as a strong load negatively affecting tested biosystems. However, the parameters of mechanical load in vitro strongly depend on the used compression equipment, hydrogels, cells, duration of compression and many other experimental conditions. Therefore, it is impossible to compare all the mechanical compression systems used in vitro with the each other or with the natural cartilage compression in vivo. In this study, the 15kPa, 1Hz, 1h/day for 7 days’ compression was negatively affecting iCa2+ and ECM systems in CS-Tyr/Gel/Cells composites, while 30 kPa, 1Hz, 1h/day for 7 days had a similar negative effect on OA cartilage explants.

            Previously we have reviewed the mechanotransductive properties of different biomimetic hydrogels and ECM components-based systems used for the cartilage tissue engineering and chondrogenic differentiation in vitro (Uzieliene et al., 2021). It is known that under physiological conditions, compressive modulus of articular cartilage varies from 0.4–2.0 MPa (Lee et al., Biomechanics of Cartilage and Osteoarthritis, 2015), which is hardly achievable in vitro. Therefore, for the hydrogels the 15 kPa increasing the release of ECM and suppressing genes expression was referred as a strong load, while the 10 kPa was positively affecting CS-Tyr/Gel/BM-MSCs by stimulating the collagen II level (the submitted another publication to IJMS). Stronger than 15kPa compression of hydrogel composites disrupted the polymerized hydrogels.

            Similar situation was observed with the OA cartilage explants, i.e. the used 30 kPa, 1 Hz, 1h/day for 7 days of mechanical load was negatively affecting OA cartilage (increased ECM COMP release, decreased expression of collagen II and aggrecan and impaired iCa2+ regulation) and was also referred as a strong load. However, the milder load (15 kPa, 1 Hz, 1h/day for 7 days) did not have a significant effect on OA cartilage. It suggests that the OA cartilage might be very sensitive to the experimental mechanical load conditions in vitro, which should be carefully experimentally chosen.

            So, the strength of mechanical load in various model systems in vitro is not the only limiting factor, i.e. the duration of compression, size and origin of cartilage explants, composition of hydrogels, used cells, media and various biocomponents should be taken into account. There is no one universal compression system in vitro. If the mechanical load worsens functioning of the chosen experimental objects (hydrogels/cells, explants and other) evaluated at ECM, intracellular protein and gene expression levels it can be named as a strong or mechanical overload, while a load positively affecting cells and cartilages – as a mild one. Following the reviewer’s suggestion, we have added such an explanation to the discussion part.  

  1. Figure 6,7: the colors representing “No TGF-β3+Load” and “With TGF-β3+Load” are too similar with each other, making it a little difficult to discriminate.  

 Answer: Thank you for the suggestion. The colors have been modified.

  1. What exactly is the relationship between iCa2+ and the down regulation of cartilage specific matrix? Is it really a causal relationship? Is it possible that another reason that truly caused the simultaneous up regulation of iCa2+ and down regulation of the cartilage specific matrix?

Answer: The numerous studies have shown that iCa2+ plays a crucial role in regulating cell functioning and cell differentiation processes [Kawano S., Shoji S., Ichinose S., Yamagata K., Tagami M., Hiraoka M. Characterization of Ca2+ signaling pathways in human mesenchymal stem cells. Cell Calcium. 2002;32:165–174. doi: 10.1016/S0143416002001240.; Viti F., Landini M., Mezzelani A., Petecchia L., Milanesi L., Scaglione S. Osteogenic differentiation of MSC through calcium signaling activation: Transcriptomics and functional analysis. PLoS ONE. 2016;11:1–21. doi: 10.1371/journal.pone.0148173.]. In addition, the role of intracellular calcium in regulation of cartilage ECM synthesis and chondrogenic differentiation of mesenchymal stem cell was reviewed in 2018 (Uzieliene et al., 2018 The Role of Physical Stimuli on Calcium Channels in Chondrogenic Differentiation of Mesenchymal Stem Cells). However, the direct relation between the iCa2+ level and chondrogenesis is still not fully clear.

            The main question - is there one mechanism by which iCa2+ can affect cartilage ECM? This is not fully elucidated so far. The iCa2+ is a potent signalling molecule in the cell, which concentration is very strictly controlled by various channels. The too high iCa2+ concentration is dreadful for the cells (activates proteases), while too low is impairing proper cell functioning. For example, an iCa2+ levels change from 100 nmol/L at “resting” state to the 1 µmol/L during the mechanical or other types of activations during the seconds or even more quicker [Clapham D.E. Calcium Signaling. Cell. 2007;131:1047–1058. doi: 10.1016/j.cell.2007.11.028.]. Another study showed that iCa2+ was increased by approximately 300%, reaching a maximal value within 50 s in the cells subjected to a hypertonic shock, following a recovery of more than 90% towards the initial [Ca2+]i within 5 min [Santches, Comp Biochem Physiol A Mol Integr Physiol. 2004 Jan;137(1):173-82. doi: 10.1016/j.cbpb.2003.09.025. The molecular mechanism of iCa2+ signalling can vary – through the L-type voltage-activated Ca2+ channels, TRPV channels or stretch-activated cation channels, activity of annexins, hyperpolarization of chondrocytes, plasma Ca(2+)-ATPase, NCE reverse mode and other. Since the iCa2+ changes very quickly and channels usually compensate each other activity, the best way to investigate the impact of iCa2+is through the cell functioning. 

            The causal relations between iCa2+ levels and ECM gene expression were demonstrated by the suppression of mechanical loading-induced up-regulation of many genes, including aggrecan, type II collagen, link proteins, c-Jun, and many MMPs by the presence of BAPTA-AM, a chelator of intracellular calcium(https://www.sciencedirect.com/science/article/pii/S0021925820670569). In addition, expression of stress protein HSP70 was significantly upregulated, while c-Jun expression was significantly suppressed during compression in the presence of BAPTA-AM. Furthermore, both of the mechanical and structural properties of the collagen fibril were shown to highly depend on the concentrations of iCa2+ ( https://www.nature.com/articles/srep46042). The mechanism of this dependence was attributed to the chelation between collagen molecules and the calcium ions. 

            In this article, we have compared the control levels of iCa2+ in two different cell types (BMMSCs and OA chondrocytes) and iCa2+ regulation during chondrogenic differentiation with/without the mechanical load. Data showed that OA chondrocytes at control level had a higher iCa2+ and also secreted a higher level of chondrogenesis important ECM, such as COMP. This study also shows that mechanical load affected iCa2+ regulation trough the channels Cav1.2 and SERCA2 and their balance, which negatively affected ECM components such as collagen II and aggrecan, particularly in OA chondrocytes. Moreover, the inverse correlation between Cav1.2, SERCA2 and cartilage ECM components collagen II, aggrecan expressions was also observed in the OA cartilages. The better iCa2+ regulation/control in BM-MSCs was related to the better their chondrogenic differentiation.

            However, based on the data obtained in this study, we can only state that impaired iCa2+ regulation through the balance of Cav1.2 and SERCA2 channel is negatively affecting chondrogenesis of both hydrogels/chondrocytes composite and OA cartilage (increase the release of ECM components and suppress cartilage gene expression). Moreover, the estimation of iCa2+ level after the long-term of cell chondrogenic differentiation or in cartilage explants is complicated since the iCa2+ measurement is usually performed microscopically or by flow cytometry in alive cells. Another solution investigating the impact of iCa2+ level in the chondrogenesis could be the application of unspecific channel blockers, siRNR, knock out mice explants or other techniques. However, even these methods might have limitations since various iCa2+ regulating channels compensate each other activity particularly during the long-term of incubations. Following the reviewer’s question, the information about possible causal relations between iCa2+ levels and ECM gene expression has been added to the Discussion part.

Reviewer 2 Report

This paper discusses the effects of mechanical loading on the chondrogenic responses of bone marrow mesenchymal stem cells and chondrocytes encapsulated in a chondroitin sulphate-based hydrogel. The authors gave a good overview of the importance of this field, performed cell studies and tried to detect the mechanical stress of the samples. The authors used a variety of methods: Cell viability and metabolic activity were checked with commercial kits, intracellular potassium was measured spectrophotometrically and with a flow cytometer, chondrogenesis was checked with the Flexcell system to generate a dynamic environment and tested with immunohistochemistry/ELISA. Cell damage is estimated spectrophotometrically, and a similar procedure was applied to COMP. RNA was extracted and RT -qPCR was used to obtain dCt values. This is a good amount of data produced for this article, but whether this is sufficient to publish in an article with such a high impact factor I cannot answer with certainty as I do not work with these techniques (so my review focuses on technical things - without plagiarism checking as I do not have good software for this). The number of cited papers is good, without excessive self-citations (4), and recent publications are mentioned in the references. The manuscript is clear, relevant to the field, and presented in a well-structured manner. 

However, here are some remarks:

For the general audience, it would be beneficial to move lines 308-323 in the introduction.

I would like to see more information about the properties of produced hydrogels - there are no references in the part about gel preparation, so the reader must assume that this is the first time this gel was made by these authors. If not, please indicate where the method was first used (and what was changed in the process), and the paper needs only minor revision for publication. If these authors are the first to make such a hydrogel, the article needs to be thoroughly revised and include information about the design of the gel, such as the general physicochemical characterization of these gels (with data such as degradation time, degree of swelling) and a comparison of these properties with similar gels. Since I believe the authors just neglected to reference the method used, this should be marked as a minor revision, but I expect this reference.

Please note that the temperature should be given as 37 °C and not 37°C. Also, room temperature should be given as 24 °C or similar, in India 28 °C is considered as room temperature. Calcium should be written as Ca2+ - superscripted - ensure this is consistent throughout the paper. The International System of Units and the standard ISO 31-0 requires a space between the number and the percent sign, which is consistent with the general practice of using an uninterrupted space between a numerical value and the corresponding unit of measurement.

Figure 1 - if the time were expressed in minutes, it would be easier to understand the data presented

There is no materials section, although the title on line 381 suggests otherwise - you should indicate where you purchased the chemicals used and explain any abbreviations (e.g., MES buffer).

Author Response

We thank the reviewer for the valuable comments and suggestions. Please find attached responses to each question and manuscript corrected accordingly. We hope the reviewer will accept our answers and corrections.

Reviewer 2:

This paper discusses the effects of mechanical loading on the chondrogenic responses of bone marrow mesenchymal stem cells and chondrocytes encapsulated in a chondroitin sulphate-based hydrogel. The authors gave a good overview of the importance of this field, performed cell studies and tried to detect the mechanical stress of the samples. The authors used a variety of methods: Cell viability and metabolic activity were checked with commercial kits, intracellular potassium was measured spectrophotometrically and with a flow cytometer, chondrogenesis was checked with the Flexcell system to generate a dynamic environment and tested with immunohistochemistry/ELISA. Cell damage is estimated spectrophotometrically, and a similar procedure was applied to COMP. RNA was extracted and RT-qPCR was used to obtain dCt values. This is a good amount of data produced for this article, but whether this is sufficient to publish in an article with such a high impact factor I cannot answer with certainty as I do not work with these techniques (so my review focuses on technical things - without plagiarism checking as I do not have good software for this). The number of cited papers is good, without excessive self-citations (4), and recent publications are mentioned in the references. The manuscript is clear, relevant to the field, and presented in a well-structured manner. 

However, here are some remarks:

For the general audience, it would be beneficial to move lines 308-323 in the introduction.

Answer: Thank you for the suggestion. The mentioned discussion part has been moved to the Introduction. We would also like to emphasize that isolation and investigation of primary cells from human cartilage and bone marrow as well as preparation and investigation of human OA cartilage explants is time, efforts and experimental conditions consuming process. In this study, we combined a partially new type of hydrogel system with two types (bone marrow and OA cartilage) tissues-derived primary cells to study the molecular mechanisms of strong mechanical load at protein, ECM and gene levels, with the particular attention to the role of iCa2+. In parallel, the molecular mechanisms of the mechanical load were confirmed in human OA cartilage explants.

Question. I would like to see more information about the properties of produced hydrogels - there are no references in the part about gel preparation, so the reader must assume that this is the first time this gel was made by these authors. If not, please indicate where the method was first used (and what was changed in the process), and the paper needs only minor revision for publication. If these authors are the first to make such a hydrogel, the article needs to be thoroughly revised and include information about the design of the gel, such as the general physicochemical characterization of these gels (with data such as degradation time, degree of swelling) and a comparison of these properties with similar gels. Since I believe the authors just neglected to reference the method used, this should be marked as a minor revision, but I expect this reference.

Answer:  We thank for the reviewer’s comment. The synthesis of CS-Tyr was done according to a previous procedure (1). The reference has been added to the Methods section.

Gelation of CS-Tyr/gelatin was mediated via di-phenolic linkage using horseradish peroxidase (HRP) enzyme and hydrogen peroxide (H2O2) (1). The mechanism has been used for various types of hydrogels. Because gelatin contains tyrosine residues, it can be crosslinked with CS-Tyr and form CS-Tyr/gelatin hydrogel. We did not find any previous study that fabricates chondroitin sulfate/gelatin hydrogel using this approach.

The swelling ratio of CS-Tyr/gelatin hydrogel based on its dried weight was about 30~35 folds. The elasticity of the hydrogels was analyzed using dynamic mechanical analysis. The additional information about the hydrogel composition, polymerization, elasticity and incorporation of the cells into the hydrogel is added to the method part. 

  1. Zhang, Y.J., et al., Injectable hydrogels from enzyme-catalyzed crosslinking as BMSCs-laden scaffold for bone repair and regeneration. Materials Science & Engineering C-Materials for Biological Applications, 2019. 96: p. 841-849.

Question. Please note that the temperature should be given as 37 °C and not 37°C. Also, room temperature should be given as 24 °C or similar, in India 28 °C is considered as room temperature. Calcium should be written as Ca2+ - superscripted - ensure this is consistent throughout the paper. The International System of Units and the standard ISO 31-0 requires a space between the number and the percent sign, which is consistent with the general practice of using an uninterrupted space between a numerical value and the corresponding unit of measurement.

Answer: Thank you for the remarks. All mentioned inaccuracies have been corrected. The additional explanation of RT has been added to the explanation of abbreviation. 

Question. Figure 1 - if the time were expressed in minutes, it would be easier to understand the data presented

Answer: Thank you for the remark. We agree that the minutes probably would be more clear, but our partners wanted to show in seconds. Hopefully, it won’t be a huge difference.

Question. There is no materials section, although the title on line 381 suggests otherwise - you should indicate where you purchased the chemicals used and explain any abbreviations (e.g., MES buffer).

Answer: The “Material” part has been deleted and only “Methods” part left. The purchasing companies were added to the “Method” part.

Round 2

Reviewer 2 Report

thank you for the answers, good luck with your future work

Author Response

Dear Reviewer,

We thank you for your suggestions and advices. We have improved the methodic part of the manuscript, as well as its results. Also, we carefully went through the paper and edited the text and English language. We hope you will find it suitable.

We highly appreciate your valuable comments.

Kind Regards,

Ilona Uzieliene
